# Riemannian Optimization for Fair Spectral Clustering

**Minh Phu Vuong** [1]   **Jinyoung Lee** [1]   **Young-Ju Lee** [2]   **Chul-Ho Lee** [1]

## Abstract

Fair graph clustering has emerged as a critical research area for addressing algorithmic bias in machine learning. The objective is to ensure that the proportion of each protected group within a cluster is consistent with its representation in the entire dataset. However, most existing spectral solutions rely on computationally expensive eigendecompositions of the graph Laplacian, limiting their scalability. In this paper, we propose Riemannian Fair Spectral Clustering (R-FairSC), a novel method that formulates fair spectral clustering as a constrained optimization problem on a Riemannian manifold. We develop a Riemannian alternating direction method of multipliers employing a variable-splitting strategy to efficiently solve the associated subproblems. Numerical experiments on large synthetic and real-world graphs demonstrate that R-FairSC significantly improves computational efficiency over state-of-the-art methods while maintaining high clustering quality and fairness.

## 1. Introduction

As machine learning systems become increasingly embedded in critical real-world applications, it is imperative to develop algorithms that maintain consistent performance across diverse demographic groups. Algorithmic decisions must not unfairly disadvantage individuals based on sensitive attributes such as race, gender, age, or socioeconomic status (Angwin et al., 2022; Farayola et al., 2023). To address these concerns, a variety of fairness-aware learning techniques have been proposed (Chierichetti et al., 2017; Ali et al., 2023; Zhang et al., 2022; Xian et al., 2023; Kleindessner et al., 2019; Wang et al., 2023; Vuong et al., 2025; Tonin et al., 2025). Among these, fair graph clustering has

emerged as a key research area for graph-structured data.

Fair graph clustering aims to partition a set of nodes into $k$ disjoint clusters such that the demographic composition of each cluster matches that of the entire population. Fair Spectral Clustering (FairSC) (Kleindessner et al., 2019) adapted the fairness principles of $k$-means clustering (Chierichetti et al., 2017) to the spectral clustering setting (Shi & Malik, 2000; Von Luxburg, 2007) by encoding fairness as linear constraints alongside the standard orthogonality constraints. However, FairSC requires computing a matrix square root, resulting in cubic-time complexity that limits its scalability. To mitigate this, Scalable FairSC (s-FairSC) (Wang et al., 2023) reformulated the problem via nullspace projection to solve a modified eigenvalue problem. Nevertheless, both methods fundamentally rely on eigensolvers, which remain computationally expensive for large graphs.

Recently, Riemannian manifold optimization (Boumal, 2023; Absil et al., 2009) has gained significant attention and has been applied across diverse domains, including power systems (Goodwin et al., 2025), resource allocation (Zargari et al., 2025), computer vision (Vemulapalli & Chellapa, 2016), robotics (Asgharivaskasi et al., 2025), and biological networks (Kan et al., 2023). Motivated by this success, we observe that fair spectral clustering can be naturally formulated as a constrained Riemannian optimization problem, where orthogonality constraints define the search space (the Stiefel manifold) and fairness imposes additional linear constraints. To the best of our knowledge, the application of Riemannian optimization to fair spectral clustering remains an unexplored research direction.

In this work, we present Riemannian Fair Spectral Clustering (R-FairSC), a scalable method that addresses the computational bottlenecks of prior eigensolver-based approaches. R-FairSC operates through three key steps. First, we reformulate fair spectral clustering as a constrained optimization problem on a Riemannian manifold, where the orthogonality constraints are encoded directly into the manifold geometry. Second, we introduce a variable-splitting strategy to decouple the fairness requirements from the manifold constraints. Third, we derive a Riemannian Alternating Direction Method of Multipliers (ADMM) to solve these subproblems via tractable, alternating updates, thereby avoiding expensive eigendecompositions.

[1]Department of Computer Science, Texas State University, San Marcos, TX, USA [2]Department of Mathematics, Texas State University, San Marcos, TX, USA. Correspondence to: Chul-Ho Lee <chulho.lee@txstate.edu>.

*Proceedings of the 43rd International Conference on Machine Learning*, Seoul, South Korea. PMLR 306, 2026. Copyright 2026 by the author(s).

Our approach is most closely related to A-FairSC (Tonin et al., 2025), which enhances scalability by reformulating fair spectral clustering as a Difference of Convex (DC) optimization problem and solving it via ADMM. However, a key distinction lies in the complexity of the resulting subproblems. While the DC framework in A-FairSC leads to computationally intensive subproblems, our variable-splitting strategy effectively decouples the fairness requirements from the manifold constraints. This decoupling yields tractable subproblems that are more amenable to efficient optimization. As demonstrated in our experimental results, this formulation allows R-FairSC to achieve superior computational efficiency compared to A-FairSC while maintaining high clustering quality.

Our contributions are threefold. (1) We propose R-FairSC, the first framework to formulate the fair spectral clustering problem as a Riemannian manifold optimization problem. (2) We develop an efficient Riemannian ADMM algorithm that utilizes variable splitting to decouple constraints, enabling scalable updates without expensive eigensolvers. (3) We extensively validate R-FairSC on large synthetic and real-world graphs, showing that it achieves superior scalability over state-of-the-art baselines without compromising clustering quality and fairness.

## 2. Preliminaries

Let $G = (\mathcal{V}, \mathcal{E})$ be an undirected weighted graph with node set $\mathcal{V} = \{1, \ldots, n\}$ and edge set $\mathcal{E}$. The graph is represented by a symmetric affinity matrix $M \in \mathbb{R}^{n \times n}$, where $m_{ij} > 0$ if $(i, j) \in \mathcal{E}$ and $m_{ij} = 0$ otherwise. We set $m_{ii} = 0$ for all $i \in \mathcal{V}$. The weighted degree of node $i$ is $d_i = \sum_{j \in \mathcal{V}} m_{ij}$, and we collect these degrees in the diagonal degree matrix $D = \text{diag}(d_1, \ldots, d_n)$. The graph Laplacian of $G$ is defined as $L = D - M$, while the normalized Laplacian is given by $\overline{L} = D^{-1/2} L D^{-1/2}$. Finally, for any two subsets $\mathcal{A}, \mathcal{B} \subseteq \mathcal{V}$, we define the link weight between them as $M(\mathcal{A}, \mathcal{B}) = \sum_{i \in \mathcal{A}, j \in \mathcal{B}} m_{ij}$.

**Notation.** We write $[n]$ for the set of integers $\{1, 2, \ldots, n\}$. For a matrix $A$, $\|A\|_F$ is the Frobenius norm and $\text{Tr}(A) = \sum_i A_{ii}$ is the trace. The Frobenius inner product between two matrices $A$ and $B$ is defined as $\langle A, B \rangle = \text{Tr}(A^\top B)$. Let $I_k$ be the $k \times k$ identity matrix. For any subset $\mathcal{A} \subseteq \mathcal{V}$, we denote its complement by $\overline{\mathcal{A}} = \mathcal{V} \setminus \mathcal{A}$.

### 2.1. Spectral clustering

The goal of graph clustering is to partition the node set $\mathcal{V}$ into $k$ disjoint sets, i.e., $\mathcal{V} = \mathcal{C}_1 \cup \mathcal{C}_2 \cup \cdots \cup \mathcal{C}_k$, by minimizing a graph cut objective. A widely adopted criterion is the normalized cut (NCut) (Von Luxburg, 2007), which quantifies the cost of partitioning a graph by measuring the total edge weight between each cluster and the remaining

vertices. It is defined as

$$\text{NCut}(\mathcal{C}_1, \mathcal{C}_2, \ldots, \mathcal{C}_k) = \frac{1}{2} \sum_{\ell=1}^{k} \frac{M(\mathcal{C}_\ell, \overline{\mathcal{C}}_\ell)}{\text{vol}(\mathcal{C}_\ell)}, \quad (1)$$

where $\text{vol}(\mathcal{C}_\ell) = \sum_{i \in \mathcal{C}_\ell} d_i$ is the volume of $\mathcal{C}_\ell$. Intuitively, minimizing NCut favors partitions where nodes assigned to the same cluster exhibit strong mutual connectivity, while nodes across different clusters have sparse connections.

For a partition $\mathcal{C}_1 \cup \cdots \cup \mathcal{C}_k$, let $V = [v_1, \ldots, v_k] \in \mathbb{R}^{n \times k}$ be the scaled indicator matrix. Each column $v_\ell \in \mathbb{R}^n$ is the scaled indicator vector for cluster $\mathcal{C}_\ell$, with entries $v_{\ell,i} = 1/\sqrt{\text{vol}(\mathcal{C}_\ell)}$ if $i \in \mathcal{C}_\ell$, and $v_{\ell,i} = 0$ otherwise. Under this definition, we observe that

$$v_\ell^\top L v_\ell = \sum_{(i,j) \in \mathcal{E}} m_{ij} (v_{\ell,i} - v_{\ell,j})^2 = \frac{M(\mathcal{C}_\ell, \overline{\mathcal{C}}_\ell)}{\text{vol}(\mathcal{C}_\ell)}. \quad (2)$$

Furthermore, by construction, the columns of $V$ satisfy the orthonormality condition $v_\ell^\top D v_\ell = \sum_{i \in \mathcal{C}_\ell} d_i / \text{vol}(\mathcal{C}_\ell) = 1$, and $v_\ell^\top D v_{\ell'} = 0$ for $\ell \neq \ell'$. This implies $V^\top D V = I_k$. Therefore, the problem of minimizing NCut can be formulated as

$$\min_{V \in \mathcal{H}} \frac{1}{2} \text{Tr}(V^\top L V) \text{ subject to } V^\top D V = I_k, \quad (3)$$

where $\mathcal{H}$ denotes the set of all valid scaled cluster indicator matrices, and $\text{Tr}(V^\top L V) = \sum_{\ell=1}^{k} v_\ell^\top L v_\ell$. However, this optimization problem is NP-hard due to the discrete nature of the set $\mathcal{H}$ (Wagner & Wagner, 1993; Von Luxburg, 2007).

To address this challenge, Shi & Malik (2000) proposed relaxing $V$ to take arbitrary real values in $\mathbb{R}^{n \times k}$, rather than restricting it to the discrete indicator form. Specifically, let $H = D^{1/2} V$. Then the constraint $V^\top D V = I_k$ becomes $H^\top H = I_k$, and the spectral clustering problem can be solved as

$$\min_{H \in \mathbb{R}^{n \times k}} \text{Tr}(H^\top \overline{L} H) \text{ subject to } H^\top H = I_k. \quad (4)$$

By the Rayleigh-Ritz theorem (Lütkepohl, 1997), the solution $H$ comprises the eigenvectors corresponding to the $k$ smallest eigenvalues of $\overline{L}$. The final cluster assignments are obtained by applying the $k$-means algorithm to the rows of $V = D^{-1/2} H$. This procedure defines the standard Spectral Clustering (SC) algorithm (Shi & Malik, 2000).

### 2.2. Fair spectral clustering

Fair graph clustering aims to partition a graph such that each cluster reflects the demographic diversity of the entire population. Suppose the node set $\mathcal{V}$ is partitioned into $h$ disjoint sensitive groups $\mathcal{V}_1 \cup \cdots \cup \mathcal{V}_h$. A partition of $\mathcal{V}$ into $k$ clusters $\mathcal{C}_1 \cup \cdots \cup \mathcal{C}_k$ is considered *fair* if the proportion

of each sensitive group within every cluster matches its representation in the entire dataset, i.e.,

$$\frac{|\mathcal{V}_s \cap \mathcal{C}_\ell|}{|\mathcal{C}_\ell|} = \frac{|\mathcal{V}_s|}{|\mathcal{V}|}, \quad \forall s \in [h], \forall \ell \in [k]. \quad (5)$$

This fairness constraint can be expressed in matrix form. Let $\hat{F} = [f_1, \ldots, f_{h-1}] \in \mathbb{R}^{n \times (h-1)}$ be the group indicator matrix, where each column $f_s \in \mathbb{R}^n$ has entries $f_{s,i} = 1_{\{i \in \mathcal{V}_s\}} - |\mathcal{V}_s|/|\mathcal{V}|$ for all $i \in [n]$ and $s \in [h-1]$. As shown in Kleindessner et al. (2019); Wang et al. (2023), the balance condition in (5) is satisfied if and only if the partition matrix $V$ satisfies

$$\hat{F}^\top V = 0_{(h-1) \times k}, \quad (6)$$

where $0_{(h-1) \times k}$ denotes the zero matrix of dimension $(h-1) \times k$. Therefore, by substituting $H = D^{1/2} V$, defining $F = D^{-1/2} \hat{F}$, and incorporating the constraint from (6) into (4), we obtain the fair spectral clustering problem:

$$\min_{H^\top H = I_k, F^\top H = 0} \text{Tr}(H^\top \overline{L} H). \quad (7)$$

## 3. Riemannian Fair Spectral Clustering

In this section, we propose R-FairSC, a scalable method for solving the fair spectral clustering problem. We reformulate the original problem as a constrained optimization problem over a Riemannian manifold. By incorporating the orthogonality constraints directly into the manifold geometry, we convert the constrained Euclidean formulation into an optimization over the manifold with linear fairness constraints. Building on this formulation, we introduce a variable-splitting strategy, leading to a Riemannian ADMM algorithm (Boyd et al., 2011). This algorithm yields tractable subproblem updates and avoids the computationally expensive eigendecomposition of the $n \times n$ Laplacian matrix, thereby enabling scalability to large graphs.

### 3.1. Riemannian manifold optimization

We briefly review the fundamentals of Riemannian manifold optimization. For an in-depth introduction, we refer the reader to Boumal (2023). Manifold optimization addresses problems of the form

$$\min_{H \in \mathcal{M}} f(H), \quad (8)$$

where $f : \mathcal{M} \to \mathbb{R}$ is a smooth objective function and $\mathcal{M}$ is a smooth manifold.

In unconstrained Euclidean optimization over $\mathbb{R}^n$, updates are typically performed by taking a step in the negative gradient direction with a step size $\alpha$ (Nocedal & Wright, 2006). To generalize this update rule to a smooth manifold $\mathcal{M}$, we must first establish the notions of gradients and search directions on $\mathcal{M}$. This generalization relies on the property

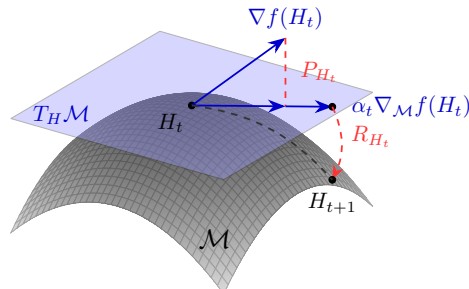

*Figure 1.* A Riemannian gradient descent step on a manifold $\mathcal{M}$.

that $\mathcal{M}$ can be locally approximated by a linear space at any point $H \in \mathcal{M}$. These local linear approximations are known as tangent spaces, denoted by $T_H \mathcal{M}$. To define a proper notion of a gradient on such a space, it is necessary to introduce an inner product on the tangent spaces. This is achieved by equipping the manifold with a Riemannian metric, which defines an inner product $\langle \cdot, \cdot \rangle_H$ on each $T_H \mathcal{M}$ that varies smoothly with $H \in \mathcal{M}$. A manifold $\mathcal{M}$ endowed with such a metric is called a Riemannian manifold.

Given any point $H \in \mathcal{M}$, the Riemannian gradient of $f$ at $H$, denoted by $\nabla_\mathcal{M} f(H)$, is the unique tangent vector in $T_H \mathcal{M}$ obtained by orthogonally projecting the standard Euclidean gradient $\nabla f(H)$ onto $T_H \mathcal{M}$ via a projection operator $P_H : \mathbb{R}^n \to T_H \mathcal{M}$. A descent step is then performed by moving from the current iterate $H_t$ along the negative gradient direction within the tangent space, scaled by a step size $\alpha_t$. Since the resulting point in the tangent space does not generally lie on the manifold, the next iterate $H_{t+1}$ is generated by mapping the point back to $\mathcal{M}$ using a retraction operator $R_{H_t} : T_{H_t} \mathcal{M} \to \mathcal{M}$. Formally, a single first-order update in Riemannian manifold optimization can be written as

$$H_{t+1} = R_{H_t} \left( -\alpha_t \nabla_\mathcal{M} f(H_t) \right). \quad (9)$$

In practice, both the projection $P_H$ and retraction $R_H$ often admit closed-form expressions. Figure 1 illustrates a descent step on a Riemannian manifold.

It is important to note that the Riemannian optimization framework can be directly applied to the $k$-way NCut in (4), since the orthogonality constraints $H^\top H = I_k$ naturally define the feasible solution set as the Stiefel manifold $\mathcal{S}(n, k) := \{H \in \mathbb{R}^{n \times k} : H^\top H = I_k\}$, which is a known Riemannian manifold (Tonin et al., 2025; Liu & Boumal, 2020). In other words, (4) becomes an unconstrained problem over $\mathcal{M} = \mathcal{S}(n, k)$, where the orthogonality constraints are enforced implicitly by the manifold geometry. Specifically, (4) can be written as

$$\min_{H \in \mathcal{S}(n,k)} f(H), \quad (10)$$

where $f(H) = \text{Tr}(H^\top \overline{L} H)$. In Appendix A.1, we empirically demonstrate that solving (10) yields clustering quality

comparable to standard SC. This observation motivates us to extend the Riemannian optimization framework to the fair graph clustering problem by introducing additional linear fairness constraints to (10). This results in a constrained optimization problem over the Stiefel manifold, given by

$$\min_{H \in \mathcal{S}(n,k)} \quad f(H)$$
$$\text{s.t.} \quad F^\top H = 0. \tag{11}$$

For simplicity, we hereafter denote $\mathcal{S}(n,k)$ simply as $\mathcal{S}$.

## 3.2. Variable-splitting strategy

The primary challenge in solving the fair graph clustering problem in (11) is to jointly satisfy the manifold constraints ($H \in \mathcal{S}$) and the fairness constraints ($F^\top H = 0$) throughout the optimization process. To address this, we propose a variable-splitting strategy that facilitates the application of the ADMM framework (Boyd et al., 2011). This approach is specifically motivated by the ability of ADMM to decouple the manifold and linear constraints, allowing them to be addressed separately and alternately. Our core idea is to introduce an auxiliary variable $Y \in \mathbb{R}^{n \times k}$ and couple it with $H$ through the consensus constraint $H - Y = 0$, leading to the following problem formulation:

$$\min_{H \in \mathcal{S}, \, Y \in \mathbb{R}^{n \times k}} \quad f(H) + g(Y)$$
$$\text{s.t.} \quad H - Y = 0, \quad F^\top Y = 0, \tag{12}$$

where we set $g(Y) = 0$. Under this framework, the manifold and fairness constraints are enforced separately within their respective subproblems. This separation ensures that each optimization step remains computationally efficient, as it avoids the complexity of satisfying both constraints simultaneously, thereby resulting in a framework that is straightforward to implement.

We next define the augmented Lagrangian for (12), which forms the basis of the ADMM updates:

$$\mathcal{L}(H, Y, U, \Lambda) = f(H) + \langle U, \, H - Y \rangle + \frac{\gamma}{2} \|H - Y\|_F^2$$
$$+ \langle \Lambda, \, F^\top Y \rangle + \frac{\eta}{2} \|F^\top Y\|_F^2, \tag{13}$$

where $U \in \mathbb{R}^{n \times k}$ and $\Lambda \in \mathbb{R}^{(h-1) \times k}$ are the dual variables associated with the coupling and fairness constraints, respectively. The parameters $\gamma > 0$ and $\eta > 0$ denote the corresponding penalty coefficients. The ADMM algorithm proceeds by iteratively solving the following subproblems:

$$H_{t+1} = \arg\min_{H \in \mathcal{S}} \mathcal{L}(H, Y_t, U_t, \Lambda_t), \tag{14}$$

$$Y_{t+1} = \arg\min_{Y \in \mathbb{R}^{n \times k}} \mathcal{L}(H_{t+1}, Y, U_t, \Lambda_t), \tag{15}$$

$$U_{t+1} = U_t + \gamma(H_{t+1} - Y_{t+1}), \tag{16}$$

$$\Lambda_{t+1} = \Lambda_t + \eta F^\top Y_{t+1}. \tag{17}$$

The optimization procedures for each subproblem are detailed in the following subsections. In essence, the ADMM algorithm alternates between minimizing the augmented Lagrangian with respect to the primal variables $(H, Y)$ and updating the dual variables $(U, \Lambda)$.

## 3.3. Riemannian optimization of the $H$-subproblem

The $H$-subproblem defined in (14) requires minimizing the augmented Lagrangian over the Stiefel manifold. Since the orthogonality constraints are intrinsic to $\mathcal{S}$, this subproblem is treated as an unconstrained Riemannian optimization problem. Among the various Riemannian solvers available (Boumal, 2023), we employ the Riemannian Conjugate Gradient (RCG) method (Sato, 2022). We select RCG because it provides a robust first-order optimization routine with proven convergence properties. In particular, its low per-iteration computational overhead is essential for scaling our framework to large graphs.

To obtain the update $H_{t+1}$, we minimize $\mathcal{L}$ with respect to $H$ while keeping the variables $(Y_t, U_t)$ fixed. By isolating the terms in (13) that depend on $H$, the optimization problem is expressed as

$$H_{t+1} = \arg\min_{H \in \mathcal{S}} f(H) + \langle U_t, H \rangle + \frac{\gamma}{2} \|H - Y_t\|_F^2, \tag{18}$$

where $f(H) = \text{Tr}(H^\top \overline{L} H)$. We employ the RCG method to solve (18) by performing iterative updates on $\mathcal{S}$ that generalize the Euclidean conjugate gradient method. By constructing search directions that are conjugate to previous ones while incorporating the negative gradient direction, this method typically achieves faster convergence than the standard steepest descent method (Sato, 2022; Shewchuk, 1994).

Specifically, each RCG iteration proceeds as follows. First, we compute the Euclidean gradient of the objective in (18) with respect to $H$. For fixed $(Y_t, U_t)$, this gradient is given by $\nabla_H \mathcal{L}(H) = 2\overline{L}H + U_t + \gamma(H - Y_t)$. This vector represents the direction of steepest ascent in the ambient Euclidean space. Next, the Riemannian gradient $\nabla_{\mathcal{S}} \mathcal{L}(H)$ is obtained by projecting the Euclidean gradient onto the tangent space $T_H \mathcal{S}$. For the Stiefel manifold, this projection admits the following closed-form expression (Boumal, 2023):

$$\nabla_{\mathcal{S}} \mathcal{L}(H) = P_H(\nabla_H \mathcal{L}) = \nabla_H \mathcal{L} - H \, \text{sym}(H^\top \nabla_H \mathcal{L}),$$

where $\text{sym}(A) = (A + A^\top)/2$.

Starting from an initial point $H_0 \in \mathcal{S}$ (initialized as the previous ADMM iterate $H_t$) and an initial search direction $d_0 = -\nabla_{\mathcal{S}} \mathcal{L}(H_0)$, we update the iterate by taking a step along the current search direction $d_\ell \in T_{H_\ell} \mathcal{S}$. Specifically, in each inner iteration $\ell$, the new iterate is computed as

$$H_{\ell+1} = R_{H_\ell}(\alpha_\ell d_\ell), \tag{19}$$

**Algorithm 1** Riemannian Conjugate Gradient on $\mathcal{S}$ (Sato, 2022)

**Input:** $\overline{L}, Y, U, \gamma > 0, H_0 \in \mathcal{S}, \tau_i, \epsilon$
**Output:** $H_{\ell^*}$

1: $\nabla_H \mathcal{L}(H_0) \leftarrow 2\overline{L}H_0 + U + \gamma(H_0 - Y)$
2: $g_0 \leftarrow P_{H_0}(\nabla_H \mathcal{L}(H_0))$
3: $d_0 \leftarrow -g_0$
4: **for** $\ell = 0, 1, \ldots, \tau_i - 1$ **do**
5:    **if** $\|g_\ell\|_F < \epsilon$ **then**
6:       **break**
7:    **end if**
8:    Compute $\alpha_\ell > 0$ satisfying sufficient decrease conditions
9:    $H_{\ell+1} \leftarrow \mathrm{qf}(H_\ell + \alpha_\ell d_\ell)$
10:   $\nabla_H \mathcal{L}(H_{\ell+1}) \leftarrow 2\overline{L}H_{\ell+1} + U + \gamma(H_{\ell+1} - Y)$
11:   $g_{\ell+1} \leftarrow P_{H_{\ell+1}}(\nabla_H \mathcal{L}(H_{\ell+1}))$
12:   $\beta_{\ell+1} \leftarrow \frac{\langle g_{\ell+1}, g_{\ell+1} \rangle}{\langle g_\ell, g_\ell \rangle}$
13:   $d_{\ell+1} \leftarrow -g_{\ell+1} + \beta_{\ell+1} \mathcal{T}_\ell(d_\ell)$
14: **end for**

where $R_{H_\ell}$ denotes the retraction operation that maps the update from the tangent space back to the manifold, and $\alpha_\ell > 0$ is a step size chosen to satisfy standard sufficient decrease conditions for the objective function (Sato, 2022).

In this work, we utilize the retraction based on the QR decomposition due to its computational efficiency, which is particularly beneficial for large graphs. The QR-based retraction is defined as $R_{H_\ell}(\alpha_\ell d_\ell) = \mathrm{qf}(H_\ell + \alpha_\ell d_\ell)$, where $\mathrm{qf}(\cdot)$ extracts the $Q$ factor of the QR decomposition. Specifically, any matrix $A \in \mathbb{R}^{n \times k}$ can be factorized as $A = QR$, where $Q \in \mathcal{S}$ and $R$ is an upper triangular $k \times k$ matrix with strictly positive diagonal elements (Absil et al., 2009).

Once the new iterate $H_{\ell+1}$ is obtained, we compute the corresponding Riemannian gradient $\nabla_\mathcal{S}\mathcal{L}(H_{\ell+1})$ by projecting the Euclidean gradient evaluated at this point onto the new tangent space $T_{H_{\ell+1}}\mathcal{S}$. Subsequently, the next conjugate search direction $d_{\ell+1}$ is constructed by combining this Riemannian gradient with the previous search direction $d_\ell$. For brevity, let $g_\ell = \nabla_\mathcal{S}\mathcal{L}(H_\ell) \in T_{H_\ell}\mathcal{S}$. Since the previous direction $d_\ell$ lies in $T_{H_\ell}\mathcal{S}$ while the new gradient $g_{\ell+1}$ belongs to the distinct tangent space $T_{H_{\ell+1}}\mathcal{S}$, it is necessary to "move" $d_\ell$ to $T_{H_{\ell+1}}\mathcal{S}$ to compute the new search direction. This is achieved via a mapping $\mathcal{T}_\ell : T_{H_\ell}\mathcal{S} \to T_{H_{\ell+1}}\mathcal{S}$, which is commonly referred to as *vector transport* (Absil et al., 2009). For the Stiefel manifold, this operation is efficiently implemented as the orthogonal projection of $d_\ell$ onto the new tangent space, given by

$$\mathcal{T}_\ell(d_\ell) = P_{H_{\ell+1}}(d_\ell) \in T_{H_{\ell+1}}\mathcal{S}.$$

The new search direction is then computed as

$$d_{\ell+1} = -g_{\ell+1} + \beta_{\ell+1}\mathcal{T}_\ell(d_\ell), \qquad (20)$$

where $\beta_{\ell+1} = \frac{\langle g_{\ell+1}, g_{\ell+1} \rangle}{\langle g_\ell, g_\ell \rangle}$ is the scalar parameter deter-

**Algorithm 2** Riemannian Fair Spectral Clustering

**Input:** $\overline{L}, D, F, k, \gamma > 0, \tau_o$
**Output:** $X \in \{0,1\}^{n \times k}$

1: $\mathcal{L}(H, Y, U) = \mathrm{Tr}(H^\top \overline{L}H) + \langle U, H-Y \rangle + \frac{\gamma}{2}\|H-Y\|_F^2$
2: Initialize $H_0 \in \mathcal{S}, Y_0 \leftarrow H_0, U_0 \leftarrow 0$
3: Compute an orthogonal basis $U_F$ of $\mathrm{col}(F)$
4: **for** $t = 0, 1, \ldots, \tau_o - 1$ **do**
5:    $H_{t+1} \leftarrow \arg\min_{H \in \mathcal{S}} \mathcal{L}(H, Y_t, U_t)$
6:    $Z \leftarrow H_{t+1} + \frac{1}{\gamma_t}U_t$
7:    $Y_{t+1} \leftarrow (I_n - U_F U_F^\top)Z$
8:    $U_{t+1} \leftarrow U_t + \gamma_t(H_{t+1} - Y_{t+1})$
9: **end for**
10: $X \leftarrow k\text{-means}(\text{rows of } D^{-1/2}H_{\tau_o})$

mined by the Fletcher-Reeves formula (Fletcher & Reeves, 1964) to ensure the convergence properties of the subproblem. This procedure repeats until a convergence condition is met, such as the norm of the Riemannian gradient falling below a tolerance $\|\nabla_\mathcal{S}\mathcal{L}(H_\ell)\|_F < \epsilon$ or reaching a maximum number of iterations. The complete RCG procedure for solving the $H$-subproblem is summarized in Algorithm 1, where $\tau_i$ denotes the maximum number of iterations, and the final iterate $H_{\ell^*}$ provides the update $H_{t+1}$ for the outer ADMM loop.

### 3.4. Solving the $Y$-subproblem

The update for the auxiliary variable $Y_{t+1}$ at iteration $t+1$ is obtained by minimizing the augmented Lagrangian $\mathcal{L}$ with respect to $Y$, keeping $H_{t+1}, U_t$, and $\Lambda_t$ fixed:

$$Y_{t+1} = \arg\min_{Y \in \mathbb{R}^{n \times k}} \mathcal{L}(H_{t+1}, Y, U_t, \Lambda_t). \qquad (21)$$

For notational simplicity, we denote $\mathcal{L}(H_{t+1}, Y, U_t, \Lambda_t)$ as $\mathcal{L}(Y)$. Unlike the $H$-subproblem defined on the Stiefel manifold, this is a minimization problem in Euclidean space. In a standard ADMM framework (14)–(17), the fairness constraints $F^\top Y = 0$ would typically be satisfied only asymptotically via the dual variable $\Lambda$ updates.

However, in this work, we enforce the fairness constraints exactly at every iteration by restricting the feasible set of $Y$ to the null space of $F^\top$, denoted as $\mathrm{null}(F^\top)$. Specifically, when $Y \in \mathrm{null}(F^\top)$, both the dual term $\langle \Lambda_t, F^\top Y \rangle$ and the penalty term $\frac{\eta}{2}\|F^\top Y\|_F^2$ vanish from (13). As a result, the $Y$-subproblem simplifies to

$$\begin{aligned} \min_{Y \in \mathbb{R}^{n \times k}} \quad & \mathcal{L}(Y), \\ \text{s.t.} \quad & Y \in \mathrm{null}(F^\top), \end{aligned} \qquad (22)$$

where $\mathcal{L}(Y) = \langle U_t, H_{t+1} - Y \rangle + \frac{\gamma}{2}\|H_{t+1} - Y\|_F^2$. Expanding $\mathcal{L}(Y)$ and collecting the terms dependent on $Y$ yields

$$\mathcal{L}(Y) = \frac{\gamma}{2}\left(-2\langle H_{t+1} + \frac{1}{\gamma}U_t, Y \rangle + \|Y\|_F^2\right) + c,$$

where $c$ represents constant terms independent of $Y$. Completing the square allows us to further rewrite this as

$$\mathcal{L}(Y) = \frac{\gamma}{2} \left\| Y - \left( H_{t+1} + \frac{1}{\gamma} U_t \right) \right\|_F^2 + c.$$

Therefore, the $Y$-subproblem in (22) reduces to

$$\min_{Y \in \mathbb{R}^{n \times k}} \quad \left\| Y - \left( H_{t+1} + \frac{1}{\gamma} U_t \right) \right\|_F^2 \qquad (23)$$
$$\text{s.t.} \qquad Y \in \text{null}(F^\top).$$

Let $Z = H_{t+1} + \frac{1}{\gamma} U_t$. Since the optimization problem in (23) corresponds to the Euclidean projection of $Z$ onto the linear subspace $\text{null}(F^\top)$, it admits a closed-form solution. Thus, the update for the auxiliary variable $Y_{t+1}$ is given by

$$Y_{t+1} = (I_n - U_F U_F^\top) Z, \qquad (24)$$

where $U_F \in \mathbb{R}^{n \times (h-1)}$ denotes an orthonormal basis for the column space $\text{col}(F)$ (Boyd & Vandenberghe, 2004).

### 3.5. Algorithm summary and complexity

We now integrate the solutions to the individual subproblems into a unified optimization routine based on the proposed variable-splitting strategy. At each ADMM iteration, the algorithm alternates between a RCG solver for the manifold-constrained $H$-subproblem and a closed-form orthogonal projection for the fairness-constrained $Y$-subproblem. This structure effectively decouples the intrinsic geometry of the Stiefel manifold from the extrinsic linear fairness constraints, facilitating efficient updates. Finally, the Lagrangian multiplier $U$ is updated via (16), where the penalty parameter $\gamma$ is replaced by a time-varying $\gamma_t$ following standard update rules (Boyd et al., 2011). The complete R-FairSC algorithm is summarized in Algorithm 2.

The total computational complexity of the R-FairSC algorithm is $O(nh^2 + \tau_o(\tau_i(|\mathcal{E}|k + nk^2) + nhk))$, where $\tau_o$ and $\tau_i$ denote the number of outer ADMM and inner RCG iterations, respectively. This complexity comprises a one-time initialization cost of $O(nh^2)$ to compute the orthonormal basis $U_F \in \mathbb{R}^{n \times (h-1)}$ for the fairness constraints, followed by the iterative ADMM updates. During the optimization phase, the per-iteration computational cost is primarily governed by the updates within the $H$ and $Y$ subproblems. For the $H$-subproblem, which is executed for $\tau_i$ inner iterations, the cost is dominated by evaluating the Euclidean gradient and performing manifold operations. Computing $\overline{L}H$ via sparse-dense matrix multiplication costs $O(|\mathcal{E}|k)$, while the tangent space projection and QR-based retraction scale as $O(nk^2)$ (Absil et al., 2009), yielding a subproblem cost of $O(\tau_i(|\mathcal{E}|k + nk^2))$. For the $Y$-subproblem, the complexity is determined by the orthogonal projection of $Z$ onto $\text{null}(F^\top)$. This is efficiently computed as

$Y_{t+1} = Z - U_F(U_F^\top Z)$ to avoid the quadratic cost of explicit $n \times n$ matrix construction. Since $U_F$ is precomputed, this projection requires only two successive matrix multiplications with a total cost of $O(nhk)$. Summing these components results in a total per-iteration complexity of $O(\tau_i(|\mathcal{E}|k + nk^2) + nhk)$.

In typical settings where $k, h \ll n$, the total complexity of R-FairSC is effectively dominated by the $O(\tau_o \tau_i |\mathcal{E}|k)$ term. This offers a distinct advantage over sFairSC, which requires solving an eigenvalue problem, a computational bottleneck that scales poorly as $n$ increases. Furthermore, while R-FairSC shares a similar theoretical complexity with A-FairSC, we empirically demonstrate in the subsequent section that R-FairSC exhibits substantially lower execution times due to the greater efficiency of our Riemannian optimization framework.

### 3.6. Convergence characteristics

To establish the convergence properties of Algorithm 2, we first observe that problem (12) can be cast within the Euclidean nonconvex ADMM framework of Magnússon et al. (2016), which studies the following problem:

$$\min_{x \in \mathcal{X}, z \in \mathcal{Z}} u(x) + v(z) \quad \text{s.t.} \quad Ax + Bz = c.$$

By identifying $x = H$, $z = Y$, $u(x) = f(H)$, $v(z) = g(Y)$, $A = I$, $B = -I$, and $c = 0$, problem (12) can be rewritten as

$$\min_{H \in \mathcal{S}, Y \in \mathcal{Y}} f(H) + g(Y) \quad \text{s.t.} \quad H - Y = 0, \qquad (25)$$

where $\mathcal{X} = \mathcal{S} = \{H \in \mathbb{R}^{n \times k} \mid H^\top H = I_k\}$ and $\mathcal{Z} = \mathcal{Y} = \{Y \in \mathbb{R}^{n \times k} \mid F^\top Y = 0\}$ denote the orthogonality and fairness constraint sets, respectively. Under this reformulation, the $H$- and $Y$-subproblems are solved exactly as described in Sections 3.3 and 3.4. Specifically, the $H$-subproblem is solved by the RCG method (Sato, 2022), while the $Y$-subproblem reduces to a Euclidean projection onto $\text{null}(F^\top)$ and therefore admits a closed-form solution.

Following the same approach as A-FairSC (Tonin et al., 2025), we can use Proposition 3 of Magnússon et al. (2016) to obtain the following convergence result.

**Proposition 3.1.** *Let $\{(H_t, Y_t, U_t)\}$ denote the sequence of primal and dual variables generated by Algorithm 2. If the dual sequence $\{U_t\}$ converges and the solution to the $H$-subproblem obtained by RCG is also a local minimizer, then every limit point of the primal sequence $\{(H_t, Y_t)\}$ satisfies the first-order necessary conditions of problem (25).*

We provide the proof of Proposition 3.1 in Appendix A.4, where we verify that problem (25) satisfies the assumptions required by Proposition 3 of Magnússon et al. (2016).

Table 1. Statistics of datasets used in the experiments

| Dataset | $|\mathcal{V}|$ | $|\mathcal{E}|$ | Sensitive Attribute | $h$ |
|---|---|---|---|---|
| Recidivism | 18,876 | 311,870 | Race | 2 |
| Credit | 29,460 | 136,196 | Education | 3 |
| Bank | 41,170 | 7,885,457 | Education | 7 |
| Diabetes | 99,492 | 150,761 | Gender | 2 |

*Remark* 3.2. Note that the $H$-subproblem shares a similar structure to the spectral clustering problem formulated in (10), where it is solved as an unconstrained Riemannian optimization problem over the Stiefel manifold, with the orthogonality constraint $H^\top H = I_k$ enforced implicitly by the manifold geometry. By applying Theorem 6.2 from Sato (2022), we can also guarantee that, under mild assumptions, the iterates generated by the RCG solver for the $H$-subproblem converge to a stationary point. While we assume this stationary point is a local minimizer in Proposition 3.1, our numerical results in Appendix A.1 corroborate the high quality of the obtained solutions. The RCG solver yields better solutions for spectral clustering than the standard eigensolver by achieving lower orthogonality errors in nearly all configurations while maintaining competitive or better NCut values across all real-world datasets.

## 4. Experiments

In this section, we present extensive experimental results to demonstrate the clustering performance and scalability of R-FairSC compared to state-of-the-art baselines, including FairSC (Kleindessner et al., 2019), sFairSC (Wang et al., 2023), A-FairSC (Tonin et al., 2025), and standard SC.

### 4.1. Experiment setup

**Datasets.** We utilize both synthetic and real-world datasets for performance evaluation. Specifically, for synthetic datasets, we employ a modified stochastic block model (mSBM) and RandLaplace graphs with up to 150,000 nodes. The mSBM (Kleindessner et al., 2019) is a standard benchmark that generates synthetic graphs with ground-truth clustering by partitioning $n$ nodes into $h$ groups and $k$ clusters, while RandLaplace (Wang et al., 2023) is a randomly generated graph used for scalability evaluation. We also consider four real-world datasets with their statistics summarized in Table 1 and details provided in Appendix A.2. For all real-world datasets, graphs are constructed by connecting nodes if the Euclidean distance between their feature vectors is smaller than a threshold (Dong et al., 2023).

**Evaluation metrics.** To evaluate clustering performance on the mSBM graphs, we use the error rate defined in Kleindessner et al. (2019). Let $X$ and $X^* \in \{0, 1\}^{n \times k}$ represent the predicted and ground-truth cluster indicator matrices, respectively. Since cluster labels are arbitrary, we assess performance by finding the optimal column permutation that aligns the predicted clustering with the ground truth.

The error rate is thus defined as

$$E(X, X^*) = \frac{1}{k} \min_{P \in \mathcal{P}_k} \|XP - X^*\|_F, \qquad (26)$$

where $\mathcal{P}_k$ denotes the set of all $k \times k$ permutation matrices.

We assess fairness on RandLaplace and real-world datasets using the average balance metric proposed in Chierichetti et al. (2017). For a dataset $\mathcal{V}$ partitioned into $h$ demographic groups $\mathcal{V}_1, \dots, \mathcal{V}_h$ and $k$ clusters $\mathcal{C}_1, \dots, \mathcal{C}_k$, the balance of an individual cluster $\mathcal{C}_l$ is defined as the minimum ratio of group representations, i.e.,

$$\text{balance}(\mathcal{C}_l) = \min_{s, s' \in [h], s \neq s'} \frac{|\mathcal{V}_s \cap \mathcal{C}_l|}{|\mathcal{V}_{s'} \cap \mathcal{C}_l|} \in [0, 1]. \qquad (27)$$

This quantifies how proportionally groups are represented within each cluster, where 1 denotes a perfectly balanced distribution and values near 0 indicate significant imbalance. We report the average balance across all $k$ clusters, with higher values indicating a fairer clustering assignment.

It is worth noting that this evaluation metric does not directly correspond to the fairness constraints $F^\top H = 0$. These group fairness constraints require that each demographic group be represented in each cluster proportionally to its overall presence in the dataset. In contrast, the balance metric in (27) evaluates within-cluster parity, i.e., how evenly demographic groups are distributed within each cluster. The average balance metric then aggregates this score across the entire partition, meaning that even if one cluster has a high balance score, poorly balanced clusters will penalize the overall average. While this evaluation metric does not perfectly match the fairness constraints in the optimization problem, it is the widely adopted standard in the literature (Kleindessner et al., 2019; Wang et al., 2023; Vuong et al., 2025; Tonin et al., 2025). Relying on this established metric ensures a consistent, direct comparison against existing baselines.

**Experimental environment.** All experiments were conducted on a Linux server equipped with an Intel Xeon Gold CPU, 95 GB of RAM, and an NVIDIA Quadro RTX 8000 GPU. The proposed framework is implemented in Python 3.12, utilizing SciPy v1.11 and Geoopt v0.5.1 (Kochurov et al., 2020)[1]. Each experiment is repeated 10 times, and the average results are reported.

**Implementation details.** For FairSC and sFairSC, we utilize the authors' publicly available source code. Since the source code for A-FairSC is unavailable, we re-implemented it following the provided pseudocode. To ensure a fair comparison with A-FairSC, which also utilizes the ADMM framework, we set the inner and outer loop iterations to $\tau_i = 200$ and $\tau_o = 50$, respectively, and initialize the penalty parameter $\gamma_0 = 0.005$ for both R-FairSC and A-FairSC.

---

[1]https://github.com/minhphuv/R-FairSC

*Table 2.* Speedup of R-FairSC over A-FairSC on mSBM

| | $n = 50,000$ | | | | $n = 100,000$ | | | | $n = 150,000$ | | | |
| --- | --- | --- | --- | --- | --- | --- | --- | --- | --- | --- | --- | --- |
| | $k=4$ | $k=5$ | $k=6$ | $k=7$ | $k=4$ | $k=5$ | $k=6$ | $k=7$ | $k=4$ | $k=5$ | $k=6$ | $k=7$ |
| $h=7$ | 3.89× | 3.95× | 3.08× | 2.21× | 2.88× | 3.39× | 4.57× | 3.51× | 4.43× | 4.31× | 4.79× | 3.97× |
| $h=8$ | 2.91× | 5.58× | 3.77× | 4.09× | 2.10× | 3.79× | 4.40× | 5.10× | 3.63× | 3.58× | 5.65× | 5.19× |
| $h=9$ | 3.86× | 4.79× | 3.69× | 4.07× | 3.33× | 4.72× | 4.81× | 3.91× | 3.57× | 4.71× | 5.53× | 5.02× |
| $h=10$ | 4.32× | 3.89× | 4.53× | 3.96× | 2.66× | 3.90× | 3.41× | 5.03× | 2.99× | 4.98× | 3.75× | 4.43× |

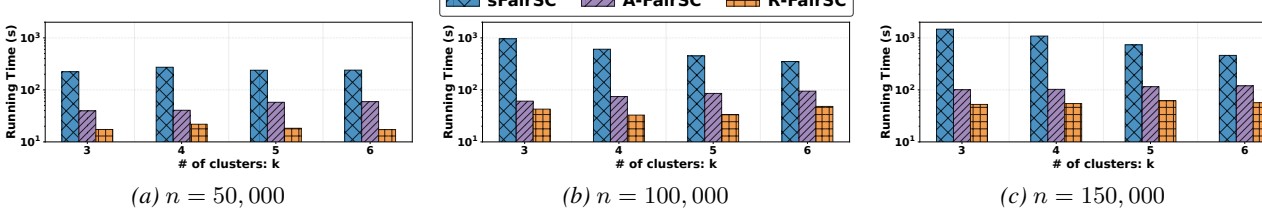

*(a) $n = 50,000$*     *(b) $n = 100,000$*     *(c) $n = 150,000$*

*Figure 2.* Runtime comparison (in seconds, log scale) on the RandLaplace dataset for varying graph sizes $n$ and cluster counts $k$.

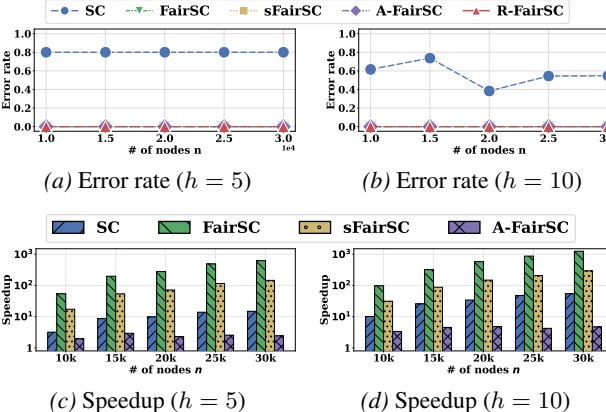

*(a) Error rate ($h = 5$)*     *(b) Error rate ($h = 10$)*

*(c) Speedup ($h = 5$)*     *(d) Speedup ($h = 10$)*

*Figure 3.* Performance comparison on mSBM ($k = 5$). The top row shows the error rate, while the bottom row depicts the computational speedup achieved by R-FairSC relative to the baselines.

### 4.2. Experimental results

**Synthetic datasets.** We first discuss experimental results on the synthetic datasets for the fair spectral clustering problem. To assess clustering performance and scalability, we evaluate R-FairSC against FairSC, sFairSC, A-FairSC, and SC in terms of error rate and runtime, starting with the mSBM dataset. As illustrated in Figure 3, R-FairSC achieves accuracy comparable to that of FairSC, sFairSC, and A-FairSC in recovering ground-truth clusters. In contrast, standard SC fails to identify the underlying cluster structure as it does not account for fairness constraints. Beyond its competitive accuracy, R-FairSC exhibits significantly enhanced scalability, yielding average speedups of $476\times$, $117\times$, and $3\times$ over FairSC, sFairSC, and A-FairSC, respectively.

Given the substantial performance lead over FairSC and sFairSC, we provide a more focused comparison with the state-of-the-art scalable baseline, A-FairSC. Here, we evaluate performance on mSBM graphs with up to 150,000 nodes and varying values of $h$ and $k$. While both methods consistently recover ground-truth clusters with zero error across all tested configurations, R-FairSC is consid-

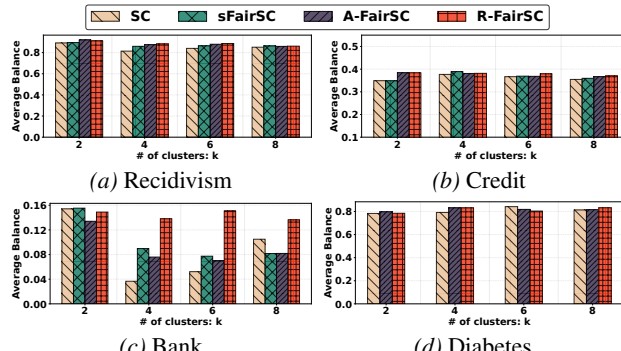

*(a) Recidivism*     *(b) Credit*

*(c) Bank*     *(d) Diabetes*

*Figure 4.* Comparison of average balance on the four real-world datasets for varying cluster counts $k$.

erably more computationally efficient. As summarized in Table 2, R-FairSC consistently outperforms A-FairSC, achieving speedups of up to $5.65\times$. Specifically, we observe average speedups of approximately $3.91\times$ and $3.84\times$ for $n = 50,000$ and $n = 100,000$, respectively, increasing to $4.25\times$ as $n$ reaches $150,000$. These results indicate that the computational advantage of R-FairSC becomes more pronounced as the problem size increases.

We further evaluate the scalability of R-FairSC against sFairSC and A-FairSC on the RandLaplace dataset. As these methods yield comparable fairness performance, we omit the balance results to focus on computational efficiency. Also, FairSC is excluded due to its prohibitive computational cost on these large-scale datasets. Figure 2 presents the runtime comparison for graph sizes ranging from $n = 50,000$ to $n = 150,000$ with varying cluster counts ($k = 3$ to 6). We observe that R-FairSC consistently achieves the lowest runtime in all settings, outperforming sFairSC and A-FairSC with average speedups of $15.2\times$ and $2.2\times$, respectively. These results confirm that R-FairSC is significantly more scalable, making it a highly practical choice for large-scale fair spectral clustering.

**Real-world datasets.** We next evaluate the effectiveness

*Table 3.* Comparison of $\|F^\top H\|_F \downarrow$ for A-FairSC and R-FairSC

| $k$ | Alg. | Recidivism | Credit | Bank | Diabetes |
|---|---|---|---|---|---|
| 2 | A-FairSC | $1.51 \times 10^{-2}$ | $7.29 \times 10^{-2}$ | $9.11 \times 10^{-2}$ | $7.10 \times 10^{-2}$ |
|   | R-FairSC | $2.07 \times 10^{-3}$ | $2.43 \times 10^{-3}$ | $5.16 \times 10^{-3}$ | $5.53 \times 10^{-5}$ |
| 4 | A-FairSC | $3.45 \times 10^{-2}$ | $1.81 \times 10^{-1}$ | $2.32 \times 10^{-1}$ | $1.41 \times 10^{-1}$ |
|   | R-FairSC | $4.98 \times 10^{-3}$ | $3.09 \times 10^{-3}$ | $5.85 \times 10^{-3}$ | $8.56 \times 10^{-5}$ |
| 6 | A-FairSC | $5.23 \times 10^{-2}$ | $2.28 \times 10^{-1}$ | $6.32 \times 10^{-1}$ | $2.22 \times 10^{-1}$ |
|   | R-FairSC | $5.55 \times 10^{-3}$ | $3.99 \times 10^{-3}$ | $8.14 \times 10^{-2}$ | $1.38 \times 10^{-4}$ |
| 8 | A-FairSC | $7.51 \times 10^{-2}$ | $2.18 \times 10^{-1}$ | $7.66 \times 10^{-1}$ | $3.12 \times 10^{-1}$ |
|   | R-FairSC | $8.41 \times 10^{-4}$ | $6.96 \times 10^{-3}$ | $2.83 \times 10^{-2}$ | $4.76 \times 10^{-4}$ |

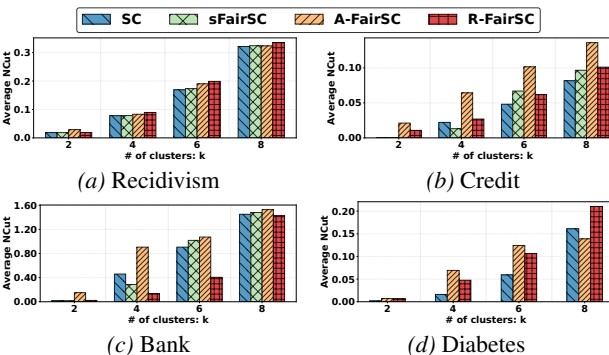

*Figure 5.* Comparison of average NCut values on the four real-world datasets for different cluster counts $k$.

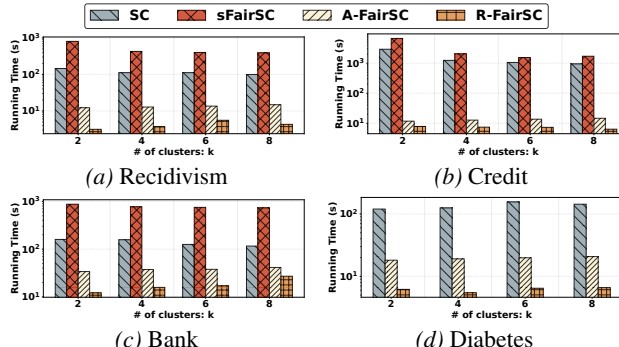

*Figure 6.* Runtime comparison (in seconds, log scale).

and efficiency of R-FairSC on four real-world datasets in terms of average balance, NCut, and runtime, comparing it against the baselines. The results are presented in Figures 4, 5, and 6. As shown in Figure 4, R-FairSC achieves average balance comparable to state-of-the-art fair clustering baselines across all datasets. In particular, on the Bank dataset, R-FairSC maintains stable balance as the number of clusters $k$ increases, whereas SC, sFairSC, and A-FairSC degrade for larger $k$. Note that we omit sFairSC results for the Diabetes dataset, as it failed to complete within our 5,000-second execution time limit. We further evaluate clustering quality via NCut values, as reported in Figure 5. Here, R-FairSC remains highly competitive, yielding up to 70% lower NCut values than A-FairSC on the Bank and Credit datasets. While we observe a marginal increase in NCut for Recidivism and Diabetes, this reflects the trade-off required to achieve higher balance under fairness constraints.

We further compare the runtime of R-FairSC with the baselines in Figure 6. R-FairSC consistently achieves the lowest runtime across all datasets, confirming its superior scalability. For instance, on the Credit dataset, R-FairSC achieves a speedup of up to $240\times$ compared to sFairSC and is over $150\times$ faster than SC. This stems from the efficiency of our Riemannian optimization framework, which avoids the high computational overhead of the eigendecomposition steps required by these baselines. Moreover, we observe a no-

ticeable improvement over A-FairSC across all datasets. Although A-FairSC also avoids eigendecomposition, it typically exhibits longer convergence time compared to R-FairSC. Overall, these results confirm that R-FairSC strikes a superior balance between fairness and efficiency, scaling reliably across all datasets while maintaining competitive clustering quality at a fraction of the computational cost.

**Fairness constraint satisfaction.** Finally, we evaluate the effectiveness of R-FairSC by comparing the constraint violations $\|F^\top H\|_F$ of A-FairSC and R-FairSC across all real-world datasets for $k \in \{2, 4, 6, 8\}$. As shown in Table 3, R-FairSC achieves lower fairness violations than A-FairSC in every setting, often by an order of magnitude or more. These results show that R-FairSC enforces the fairness constraints more strictly than A-FairSC in practice.

## 5. Conclusion

In this work, we developed R-FairSC, which leverages Riemannian manifold optimization to overcome the scalability limitations of existing fair spectral clustering methods. Our Riemannian ADMM algorithm, utilizing variable splitting, yields tractable subproblems amenable to efficient updates, making it applicable to large graphs. Experimental results across diverse datasets validate that R-FairSC maintains competitive clustering quality and fairness while achieving substantial improvements in computational efficiency.

## Acknowledgments

This work was supported by the National Science Foundation under grants IIS-2209921, CNS-2209922, and DMS-2208499.

## Impact Statement

This paper aims to advance the field of machine learning by developing a scalable fair spectral clustering algorithm. Our work has the potential for positive societal impact by enabling the broader adoption of fairness-aware clustering in real-world applications involving large-scale datasets. We do not anticipate any specific negative societal consequences that need to be highlighted here.

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

# A. Appendix

## A.1. Riemannian manifold optimization for spectral clustering

In this subsection, we demonstrate the effectiveness of Riemannian manifold optimization for solving the standard spectral clustering problem (4) by comparing it with standard SC. Specifically, we assess clustering quality on the Credit, Recidivism, Bank, and Diabetes datasets using three primary metrics, namely, the NCut value defined in (1) to measure partition quality, the orthogonality error $\|H^\top H - I_k\|_F$ to quantify how well the solution $H$ satisfies the constraint, and the clustering objective $\mathrm{Tr}(H^\top \overline{L} H)$ in (4). As shown in Table 4, Riemannian optimization (RO) achieves results comparable to SC. In particular, RO consistently yields orthogonality errors that are one to two orders of magnitude lower than those of SC for almost all datasets and cluster counts, demonstrating stricter adherence to the orthogonality constraint. Furthermore, RO frequently outperforms SC in terms of partition quality. For instance, it achieves lower NCut values for $k = 6, 8, 10$ on the Credit dataset and for $k = 10$ on the Bank dataset. These results confirm that Riemannian optimization serves as a robust alternative to standard eigendecomposition for the SC problem, often yielding numerically superior solutions. In Section 4, we demonstrate that this approach is even more advantageous for the fair spectral clustering problem.

*Table 4.* Performance comparison of clustering without fairness constraints. Ortho. indicates the orthogonal constraint and Obj. indicates the clustering objective.

*(a)* Credit

| $k$ | Alg. | NCut ↓ | Ortho. ↓ | Obj. ↓ |
|---|---|---|---|---|
| 4 | SC | 0.0184 | 9.38e-16 | 0.0184 |
|   | RO | 0.0242 | 1.08e-15 | 0.0254 |
| 6 | SC | 0.0503 | 1.13e-15 | 0.0503 |
|   | RO | 0.0466 | 1.11e-15 | 0.0483 |
| 8 | SC | 0.0848 | 1.25e-15 | 0.0848 |
|   | RO | 0.0804 | 8.42e-16 | 0.0914 |
| 10 | SC | 0.1289 | 8.92e-14 | 0.1289 |
|   | RO | 0.1196 | 1.20e-15 | 0.1236 |

*(b)* Recidivism

| $k$ | Alg. | NCut ↓ | Ortho. ↓ | Obj. ↓ |
|---|---|---|---|---|
| 4 | SC | 0.0748 | 1.24e-14 | 0.0065 |
|   | RO | 0.0748 | 6.78e-16 | 0.0065 |
| 6 | SC | 0.1903 | 1.16e-14 | 0.0213 |
|   | RO | 0.1923 | 7.88e-16 | 0.0214 |
| 8 | SC | 0.3202 | 1.05e-14 | 0.0441 |
|   | RO | 0.3216 | 9.86e-16 | 0.0442 |
| 10 | SC | 0.5781 | 1.36e-14 | 0.0814 |
|   | RO | 0.5784 | 1.09e-15 | 0.0815 |

*(c)* Bank

| $k$ | Alg. | NCut ↓ | Ortho. ↓ | Obj. ↓ |
|---|---|---|---|---|
| 4 | SC | 0.0587 | 4.75e-15 | 0.0452 |
|   | RO | 0.0573 | 9.16e-16 | 0.0454 |
| 6 | SC | 0.3858 | 6.92e-15 | 0.1001 |
|   | RO | 0.3858 | 1.10e-15 | 0.1001 |
| 8 | SC | 1.3992 | 7.53e-15 | 0.2093 |
|   | RO | 1.3998 | 1.38e-15 | 0.2093 |
| 10 | SC | 2.1960 | 8.05e-15 | 0.3928 |
|   | RO | 2.1079 | 1.45e-15 | 0.3937 |

*(d)* Diabetes

| $k$ | Alg. | NCut ↓ | Ortho. ↓ | Obj. ↓ |
|---|---|---|---|---|
| 4 | SC | 0.0439 | 1.02e-14 | 0.0057 |
|   | RO | 0.0467 | 8.10e-16 | 0.0058 |
| 6 | SC | 0.0966 | 1.72e-14 | 0.0141 |
|   | RO | 0.1022 | 9.39e-16 | 0.0144 |
| 8 | SC | 0.2093 | 2.00e-14 | 0.0248 |
|   | RO | 0.2114 | 1.02e-15 | 0.0250 |
| 10 | SC | 0.3611 | 2.57e-14 | 0.0365 |
|   | RO | 0.3961 | 1.38e-15 | 0.0366 |

## A.2. Details of the datasets

Here, we provide further details on the four benchmark datasets used in our experiments. The Recidivism dataset (Jordan & Freiburger, 2015) comprises U.S. state-court defendant records (1990–2009) for individuals released on bail, with race treated as the protected attribute. The Credit dataset (Yeh & Lien, 2009) features credit-card client records, with education as the sensitive attribute. The Bank Marketing dataset (Moro et al., 2014) consists of direct marketing records from phone-call campaigns of a Portuguese bank, where education serves as the sensitive attribute. Finally, the Diabetes dataset (Clore et al., 2014) contains inpatient encounter records for diabetic patients across 130 U.S. hospitals (1999–2008), with gender as the sensitive attribute.

## A.3. Additional experimental results

We first provide additional results for the runtime of the $H$-subproblem for A-FairSC and R-FairSC, as this subproblem dominates the total execution time of both algorithms. To ensure a fair comparison, we set the number of iterations for the $H$-subproblem to be identical for both methods. As shown in Table 5, R-FairSC is substantially faster across all test cases, achieving an overall average runtime of 9.82 seconds compared to 14.93 seconds for A-FairSC. This advantage is most notable on the Bank dataset, where R-FairSC reduces the $H$-update time by about $40\%$, confirming the superior efficiency and scalability of our approach. These results empirically validate that our variable-splitting strategy yields subproblems that are computationally more efficient to solve within the Riemannian optimization framework.

*Table 5.* Comparison of $H$-subproblem runtimes (in seconds) for A-FairSC and R-FairSC

| $k$ | Alg. | Credit | Recidivism | Bank | Diabetes |
|---|---|---|---|---|---|
| 4 | A-FairSC | 6.60 | 6.57 | 30.97 | 6.83 |
|   | R-FairSC | 6.59 | 5.43 | 14.60 | 4.86 |
| 6 | A-FairSC | 8.14 | 7.62 | 31.59 | 8.20 |
|   | R-FairSC | 6.47 | 6.37 | 15.69 | 5.12 |
| 8 | A-FairSC | 9.09 | 8.80 | 35.05 | 9.94 |
|   | R-FairSC | 5.83 | 7.67 | 25.79 | 4.48 |
| 10 | A-FairSC | 10.44 | 10.13 | 36.48 | 12.47 |
|   | R-FairSC | 6.12 | 7.88 | 28.79 | 5.38 |

Next, we provide the standard deviations across all metrics for all real-world datasets in Tables 6, 7, and 8. Overall, R-FairSC achieves lower variance than A-FairSC in the majority of settings. This confirms that R-FairSC not only performs better on average but also yields higher solution stability.

*Table 6.* Comparison of Balance↑ for A-FairSC and R-FairSC

| $k$ | Alg. | Recidivism | Credit | Bank | Diabetes |
|---|---|---|---|---|---|
| 2 | A-FairSC | $0.9194 \pm 0.0238$ | $0.3840 \pm 0.0012$ | $0.1341 \pm 0.0137$ | $0.7967 \pm 0.0209$ |
|   | R-FairSC | $0.9100 \pm 0.0016$ | $0.3838 \pm 0.0012$ | $0.1490 \pm 0.0000$ | $0.7846 \pm 0.0000$ |
| 4 | A-FairSC | $0.8753 \pm 0.0236$ | $0.3810 \pm 0.0036$ | $0.0760 \pm 0.0196$ | $0.8308 \pm 0.0412$ |
|   | R-FairSC | $0.8818 \pm 0.0063$ | $0.3816 \pm 0.0064$ | $0.1380 \pm 0.0002$ | $0.8310 \pm 0.0100$ |
| 6 | A-FairSC | $0.8796 \pm 0.0148$ | $0.3685 \pm 0.0106$ | $0.0702 \pm 0.0004$ | $0.8182 \pm 0.0255$ |
|   | R-FairSC | $0.8860 \pm 0.0007$ | $0.3795 \pm 0.0033$ | $0.1508 \pm 0.0000$ | $0.8040 \pm 0.0290$ |
| 8 | A-FairSC | $0.8592 \pm 0.0089$ | $0.3667 \pm 0.0079$ | $0.0818 \pm 0.0048$ | $0.8150 \pm 0.0203$ |
|   | R-FairSC | $0.8614 \pm 0.0044$ | $0.3701 \pm 0.0073$ | $0.1363 \pm 0.0000$ | $0.8318 \pm 0.0056$ |

*Table 7.* Comparison of NCut↓ for A-FairSC and R-FairSC

| $k$ | Alg. | Recidivism | Credit | Bank | Diabetes |
|---|---|---|---|---|---|
| 2 | A-FairSC | $0.0291 \pm 0.0085$ | $0.0212 \pm 0.0053$ | $0.1501 \pm 0.1689$ | $0.0071 \pm 0.0089$ |
|   | R-FairSC | $0.0190 \pm 0.0011$ | $0.0107 \pm 0.0058$ | $0.0218 \pm 0.0000$ | $0.0067 \pm 0.0000$ |
| 4 | A-FairSC | $0.0832 \pm 0.0064$ | $0.0643 \pm 0.0118$ | $0.9060 \pm 0.1109$ | $0.0692 \pm 0.0172$ |
|   | R-FairSC | $0.0894 \pm 0.0024$ | $0.0269 \pm 0.0039$ | $0.1373 \pm 0.0011$ | $0.0476 \pm 0.0061$ |
| 6 | A-FairSC | $0.1901 \pm 0.0274$ | $0.1015 \pm 0.0168$ | $1.0734 \pm 0.0111$ | $0.1243 \pm 0.0538$ |
|   | R-FairSC | $0.1987 \pm 0.0007$ | $0.0618 \pm 0.0038$ | $0.4057 \pm 0.0000$ | $0.1066 \pm 0.0122$ |
| 8 | A-FairSC | $0.3235 \pm 0.0053$ | $0.1360 \pm 0.0150$ | $1.5280 \pm 0.0102$ | $0.1389 \pm 0.0639$ |
|   | R-FairSC | $0.3349 \pm 0.0438$ | $0.1010 \pm 0.0087$ | $1.4296 \pm 0.0018$ | $0.2104 \pm 0.0304$ |

*Table 8.* Comparison of Runtime↓ for A-FairSC and R-FairSC

| $k$ | Alg. | Recidivism | Credit | Bank | Diabetes |
|---|---|---|---|---|---|
| 2 | A-FairSC | $11.16 \pm 0.22$ | $11.77 \pm 0.22$ | $33.92 \pm 1.05$ | $18.17 \pm 0.34$ |
|   | R-FairSC | $3.14 \pm 0.40$ | $7.87 \pm 1.74$ | $12.17 \pm 0.64$ | $6.19 \pm 1.63$ |
| 4 | A-FairSC | $11.64 \pm 0.21$ | $12.77 \pm 0.25$ | $37.39 \pm 1.72$ | $19.06 \pm 0.15$ |
|   | R-FairSC | $3.64 \pm 0.21$ | $7.45 \pm 1.34$ | $15.81 \pm 1.45$ | $5.47 \pm 0.52$ |
| 6 | A-FairSC | $12.24 \pm 0.27$ | $13.69 \pm 0.25$ | $37.81 \pm 0.72$ | $19.84 \pm 0.24$ |
|   | R-FairSC | $5.56 \pm 0.43$ | $7.34 \pm 0.47$ | $16.93 \pm 1.46$ | $6.46 \pm 1.31$ |
| 8 | A-FairSC | $13.23 \pm 0.23$ | $14.69 \pm 0.13$ | $41.35 \pm 2.69$ | $20.74 \pm 0.32$ |
|   | R-FairSC | $4.29 \pm 0.30$ | $6.38 \pm 0.35$ | $27.13 \pm 1.78$ | $6.61 \pm 1.19$ |

### A.4. Proof of Proposition 3.1

*Proof.* To prove Proposition 3.1, we verify below that problem (25) satisfies the following assumptions, which allow us to apply Proposition 3 from Magnússon et al. (2016).

**Assumption A.1** (Smoothness)**.** The functions $f$ and $g$ are continuously differentiable.

The function $f(H) = \mathrm{Tr}(H^\top \overline{L} H)$ is quadratic in $H$, so it is continuously differentiable. Since $g(Y) \equiv 0$ is a constant function, it is also continuously differentiable. Thus, Assumption A.1 holds.

**Assumption A.2** (Finite smooth representation of the constraint sets)**.** The constraint sets $\mathcal{S}$ and $\mathcal{Y}$ are closed and can be expressed in terms of a finite number of equality and inequality constraints.

Proposition 3 of Magnússon et al. (2016) requires each constraint set to be closed and to admit a finite representation of the form

$$\{x \mid \psi(x) = 0, \ \phi(x) \leq 0\},$$

where $\psi$ and $\phi$ have finitely many continuously differentiable components.

We first verify these properties for $\mathcal{S}$. Let $\{H^{(r)}\}_{r \geq 1}$ be any sequence in $\mathcal{S}$ such that $H^{(r)} \to H^*$. Since $H^{(r)} \in \mathcal{S}$ for every $r$, we have

$$(H^{(r)})^\top H^{(r)} = I_k.$$

Taking the limit as $r \to \infty$ and applying the continuity of matrix multiplication gives

$$\lim_{r \to \infty} \left( (H^{(r)})^\top H^{(r)} \right) = \left( \lim_{r \to \infty} (H^{(r)})^\top \right) \left( \lim_{r \to \infty} H^{(r)} \right) = (H^*)^\top H^* = I_k.$$

Thus, $H^* \in \mathcal{S}$, and hence $\mathcal{S}$ is closed.

It remains to verify that $\mathcal{S}$ admits a finite smooth representation. By Lemma D.3 of A-FairSC (Tonin et al., 2025), the constraint $H^\top H = I_k$ can be expressed as $k(k+1)/2$ scalar equations:

$$\psi_{ij}(H) = \sum_{p=1}^{n} h_{pi} h_{pj} - \delta_{ij} = 0, \quad 1 \leq i \leq j \leq k,$$

where $\delta_{ij} = 1$ if $i = j$, and $\delta_{ij} = 0$ otherwise. Since each component $\psi_{ij}$ is a quadratic polynomial in the entries of $H$, the collection $\psi = (\psi_{ij})$ is smooth and continuously differentiable.

Using a similar argument, we can verify these properties for $\mathcal{Y}$. In particular, the set $\mathcal{Y}$ is closed by the same sequential limit argument. For the finite smooth representation, the constraint $F^\top Y = 0$ yields $(h-1)k$ scalar linear equations, each of which is trivially continuously differentiable.

Since both $\mathcal{S}$ and $\mathcal{Y}$ are closed and can be expressed through finitely many smooth equality constraints, Assumption A.2 is satisfied.

**Assumption A.3** (Optimality of the ADMM subproblem solutions)**.** At every iteration of Algorithm 2, the computed solutions of the $H$-subproblem and the $Y$-subproblem are locally or globally optimal.

Since the $Y$-subproblem admits a closed-form solution given by the Euclidean projection onto $\mathrm{null}(F^\top)$, it is globally optimal. For the $H$-subproblem, the local optimality condition is imposed in Proposition 3.1, which requires the solution generated by the RCG solver to be a local minimizer. Hence, Assumption A.3 holds.

**Assumption A.4** (Linear independence constraint qualification). At every limit point generated by Algorithm 2, the constraint gradients associated with $\mathcal{S}$ and $\mathcal{Y}$ satisfy the linear independence constraint qualification.

This assumption requires that at every limit point, the gradients of the active constraints defining $\mathcal{S}$ and $\mathcal{Y}$ are linearly independent. Because the constraints defining $\mathcal{S}$ and $\mathcal{Y}$ involve disjoint sets of variables ($H$ and $Y$, respectively), it suffices to verify the constraint qualifications for $\mathcal{S}$ and $\mathcal{Y}$ individually. For $\mathcal{S}$, by Lemma D.6 of A-FairSC (Tonin et al., 2025), the gradients of the constraints defining $H^\top H = I_k$ are linearly independent at every $H \in \mathcal{S}$. Similarly for $\mathcal{Y}$, Lemma D.7 of A-FairSC (Tonin et al., 2025) demonstrates that the gradients of the constraints defining $F^\top Y = 0$ are linearly independent whenever $F$ has full column rank. Following Wang et al. (2023), this full-rank condition is guaranteed by constructing $F$ with $h-1$ columns, consistent with our setup. Therefore, the linear independence constraint qualification conditions hold at every limit point, and Assumption A.4 is satisfied. $\qquad\square$

