# OpenReview forum: "Riemannian Optimization for Fair Spectral Clustering"
_ICML.cc/2026/Conference — ICML 2026 regular_

### Official Review · Reviewer_b7q8 · 2026-03-07

**Soundness:** 3
**Presentation:** 4
**Significance:** 2
**Originality:** 3
**Overall Recommendation:** 4
**Confidence:** 4

**Summary:**

This paper introduces a Riemannian manifold optimization algoritm to address the fair spectral clustering problem. The authors pose the fair spectral clustering problem as a constrained optimization over a Riemannian manifold, i.e., the Stiefel manifolde. A Riemannian ADMM algorithm, utilizing variable splitting, is introduced for scalable solutiions to the problem. The proposed method is evaluated on both synthetic and real world data with varying number of nodes and number of groups. The evaluation is performed with respect to run time, clustering error and balance of the resulting clusters. The method is compared with respect to well-known fair spectral clustering methods.

**Compliance With Llm Reviewing Policy:**

Affirmed.

**Final Justification:**

I think this is a valuable contribution to the conference. The rebuttal addressed my my main concerns. My original evaluation was already positive so I am not changing it.

**Key Questions For Authors:**

Why are standard metrics such as NMI or ARI not used for evaluating the clustering methods?

**Limitations:**

Yes.

**Strengths And Weaknesses:**

- Soundness: The paper is technically sound.
- Presentation: The paper is very well-written and easy to follow. The authors introduce all of the necessary background which makes it easy to follow the paper.
- Significance: The paper address the problem of scalability in fair spectral clustering. This would advance our capabilities for dealing with large networks. The results show a clear impact on computational complexity. However, the impact on improving both the clustering performance and the balance of the resulting partitions is not as clear. Furthermore, the improvements in performance are either minimal with respect to A-FairSC. The evaluation of the method is a bit limited as it is not clear how robust the proposed method is to missing labels or other perturbations.
- Originality: While the paper makes a clear contribution to the scaliability of FairSC, the methods used for implementing the optimization are not particularly novel. The use of Riemannian manifolds for solving the problem is innovative even though existing methods to solve it are used.

---

> ### Author Rebuttal · Authors · 2026-03-31
>
> We thank the reviewer for the constructive feedback. Below we respond to the concerns raised.
>
> > #### W1. The impact on improving both the clustering performance and the balance of the resulting partitions is not as clear. Furthermore, the improvements in performance are either minimal with respect to A-FairSC. The evaluation of the method is a bit limited as it is not clear how robust the proposed method is to missing labels or other perturbations.
>
> Thank you for your comment. We would like to emphasize that the primary objective of this work is to address the scalability limitations of fair spectral clustering, rather than to uniformly improve clustering quality metrics across all baselines. As demonstrated in our experiments, R-FairSC maintains highly competitive clustering quality and comparable average balance to state-of-the-art methods like A-FairSC, while delivering substantial computational gains. On synthetic datasets, we observe speedups of up to $5.65\times$ over A-FairSC. Specifically, on mSBM, the average speedups are $3.91\times$, $3.84\times$, and $4.25\times$ at $n=50{,}000$, $100{,}000$, and $150{,}000$, respectively, demonstrating that our computational advantage becomes even more pronounced as problem size increases. On real-world datasets, we similarly observe clear runtime improvements with speedups up to $3.48\times$.
>
> We agree that robustness to missing sensitive labels and network perturbations is an important practical issue. However, addressing incomplete or noisy attributes requires fundamentally different problem formulations. As our focus is establishing a scalable framework for fair spectral clustering, we leave these robust extensions for future work.
>
> ---
>
> > #### W2. While the paper makes a clear contribution to the scalability of FairSC, the methods used for implementing the optimization are not particularly novel. The use of Riemannian manifolds for solving the problem is innovative even though existing methods to solve it are used.
>
> Thank you for recognizing the innovative nature of our approach. We agree that the underlying Riemannian solvers we utilize are established tools. However, the core technical novelty and non-trivial contribution of our work lie in enabling their application through a novel problem formulation. In the standard formulation, the variable $H$ must simultaneously satisfy two fundamentally different types of constraints: the orthogonality constraint and the linear fairness constraints. This coupled structure makes the problem difficult to solve directly. Our key idea is a variable-splitting approach that introduces an auxiliary variable $Y$ to explicitly handle the fairness constraints, leaving the main variable $H$ subject only to the orthogonality constraint. This decoupling makes the resulting subproblems much easier to handle. The $H$-subproblem can be optimized efficiently on the Stiefel manifold using Riemannian optimization, while the $Y$-subproblem admits a closed-form solution. As a result, our method avoids the computationally expensive eigendecomposition required by prior fair spectral clustering methods, achieving superior computational efficiency while maintaining high clustering quality.
>
> ---
>
> > #### Q1. Why are standard metrics such as NMI or ARI not used for evaluating the clustering methods?
>
> Thank you for the question. We would like to clarify that metrics such as NMI and ARI rely on ground-truth class labels to measure how closely an obtained clustering matches those predefined categories. However, the objective of fair spectral clustering is not to recover a pre-existing class partition, but rather to find graph partitions that simultaneously achieve high clustering quality and satisfy fairness constraints. In particular, for the real-world datasets considered in this paper, there are no ground-truth partitions known to inherently satisfy these fairness constraints. Thus, even when class labels are available, they do not represent a fair graph partition and therefore do not provide an appropriate reference for evaluating our specific objective. For this reason, we rely on NCut and average balance as the most appropriate evaluation metrics, since they directly measure the two quantities of interest: the structural (clustering) quality and the fairness of the resulting partitions.

---

> > ### Author Rebuttal · Reviewer_b7q8 · 2026-04-01
> >
> > The authors have addressed my concerns and clarified my questions adequately.

---

### Official Review · Reviewer_rPpY · 2026-03-11

**Soundness:** 2
**Presentation:** 3
**Significance:** 3
**Originality:** 3
**Overall Recommendation:** 4
**Confidence:** 3

**Summary:**

This paper studies fair spectral clustering under group fairness constraints, where each cluster should reflect the global protected-group proportions. The authors reformulate the relaxed fair spectral clustering problem as optimization over the Stiefel manifold with additional linear fairness constraints, then introduce a variable-splitting scheme and a Riemannian ADMM-style algorithm, R-FairSC. Empirically, the method is compared against FairSC, s-FairSC, A-FairSC, and standard spectral clustering on synthetic and real-world graphs, with the main reported finding being substantially improved runtime while maintaining competitive clustering quality and fairness.

**Compliance With Llm Reviewing Policy:**

Affirmed.

**Final Justification:**

My major concerns were addressed.

**Key Questions For Authors:**

See weakness

**Limitations:**

Yes

**Strengths And Weaknesses:**

**Strengths:**

1. The problem is relevant and well motivated. Fair spectral clustering is a meaningful area, and improving scalability is important if these methods are to be usable beyond toy graphs.
2. The formulation is clean. The progression from the fair spectral relaxation in Equation (7) to the split formulation in Equation (12) is easy to follow, and the projection solution in Equation (23) is simple and practically appealing.
3. The paper includes the right baselines for its immediate problem setting. Comparing against FairSC, sFairSC, A-FairSC, and SC is the correct starting point.
4. The runtime evidence is consistent and not cherry-picked. Figure 2, Figure 6, and Table 2 all point in the same direction, namely that R-FairSC is faster than prior fair spectral clustering baselines across several settings.

**Weaknesses:**

1. The paper does not convincingly show that the returned solution satisfies the fairness constraint it claims to optimize.
2. The fairness metric used in experiments is misaligned with the fairness definition in the method.
3. There is no convergence analysis or formal guarantee for the proposed alternating scheme.
4. The synthetic quality benchmark is too easy and largely saturated. In Figure 3 and Table 2 on Page 7, all fair methods essentially achieve zero error on mSBM. That means the benchmark no longer discriminates between methods on clustering quality. Once every fair baseline is perfect, the experiment mostly becomes a runtime comparison. That is fine for one claim, but it is weak evidence for the stronger claim that the method preserves clustering quality in challenging settings.
5. The analysis lacks variance reporting and deeper diagnostics.

---

> ### Author Rebuttal · Authors · 2026-03-31
>
> We thank the reviewer for the constructive feedback. Below we respond to the concerns raised.
>
> > #### Regarding W1.
>
> Thank you for the comment. To demonstrate that our solution satisfies the fairness constraints, we evaluated the constraint violation $\|F^\top H\|_F$. Across all real-world datasets and $k \in \\{2,4,6,8\\}$, R-FairSC consistently achieves lower violations (ranging from $5.53\times10^{-5}$ to $8.14\times10^{-2}$) than A-FairSC (ranging from $1.51\times10^{-2}$ to $7.66\times10^{-1}$). This confirms that our method successfully enforces the constraints. We will include complete results in the revised manuscript.
>
> ---
>
> > #### Regarding W2.
>
> Thank you for the comment. We agree that the distinction between our method's fairness formulation and the empirical evaluation metric should be clarified. Our method enforces the standard group fairness constraints, which requires that each demographic group be represented in each cluster proportionally to its overall presence in the dataset. In contrast, the balance metric in Eq. (25) evaluates within-cluster parity, i.e., how evenly demographic groups are distributed within each cluster. The average balance metric then aggregates this score across the entire partition (meaning that even if one cluster has a high balance score, poorly balanced clusters will penalize the overall average). While this evaluation metric does not perfectly match the fairness constraints in the optimization problem, it is the widely adopted in the literature. Relying on this established metric ensures a fair, direct comparison against existing baselines. We will explicitly clarify this distinction in the revised manuscript.
>
> ---
>
> > #### Regarding W3.
>
> Thank you for raising this important point. While we formulate the fair spectral clustering problem as a constrained optimization problem on a Riemannian manifold, it can be equivalently cast within the Euclidean nonconvex ADMM framework of [1]. Specifically, we can rewrite problem (12) as:
> $$
> \min_{H\in\mathcal{S}, Y\in\mathcal{Y}} f(H)+g(Y)\quad\text{s.t.}\quad H-Y=0,
> $$
> where $\mathcal{S}=\\{H\in\mathbb{R}^{n\times k}:H^\top H=I_k\\}$ and $\mathcal{Y}=\\{Y\in\mathbb{R}^{n\times k}:F^\top Y=0\\}$ denote the orthogonality and fairness constraint sets, respectively. Under this reformulation, the $H$- and $Y$-subproblems are solved exactly as described in our paper. Specifically, the $H$-subproblem is solved by the RCG method, while the $Y$-subproblem reduces to a Euclidean projection onto $\mathrm{null}(F^\top)$ and therefore admits a closed-form solution.
>
> We note that under the assumptions of Theorem 6.2 in [Sato, 2022], the iterates generated by the RCG method for the $H$-subproblem converge to a stationary point. Assuming this stationary point is a local minimizer, we can follow the exact same argument as in Proposition 3 of [1], which A-FairSC also adapts to establish convergence, to characterize the convergence of R-FairSC. Specifically, if the dual sequence (the $U$ sequence) converges, then every limit point of the primal $(H,Y)$ sequence satisfies the first-order necessary conditions of this reformulated problem. We will discuss the convergence characteristics of R-FairSC in the revised manuscript.
>
> [1] S. Magnússon et al. "On the convergence of alternating direction Lagrangian methods for nonconvex structured optimization problems," IEEE TCNS, 2015.
>
> ---
>
> > #### Regarding W4.
>
> We agree that the mSBM benchmark is close to saturation in terms of clustering error, making it less discriminative for comparing clustering quality among state-of-the-art methods. However, we believe these experiments remain important. First, mSBM is the standard synthetic graph generator used in the literature, allowing for a direct and expected comparison with prior work. Second, it provides flexible synthetic settings for controlled performance evaluation under different fairness and clustering configurations, while providing exact ground-truth labels for quality evaluation. Therefore, the experiments on mSBM confirm that R-FairSC perfectly matches the optimal clustering quality of prior methods in controlled settings while offering superior computational efficiency. We also note that we demonstrate superior performance of R-FairSC on real-world datasets, which complement the experiments on mSBM
>
> ---
>
> > #### Regarding W5.
>
> Due to the space limit, we provide a summary of the standard deviations across all real-world datasets. Specifically, the standard deviation for R-FairSC ranges from $0$ to $0.029$ in average balance (vs. $0.0004$ to $0.0412$ for A-FairSC), from $0$ to $0.0438$ in NCut (vs. $0.0053$ to $0.1689$), and from $0.21$ to $1.78$ seconds in runtime (vs. $0.13$ to $2.69$ seconds). Overall, R-FairSC shows lower variance than A-FairSC in the majority settings. This confirms that R-FairSC not only performs better on average but also yields higher solution stability. We will include the complete results in the revised manuscript.

---

> > ### Author Rebuttal · Reviewer_rPpY · 2026-04-03
> >
> > Thank you for the detailed rebuttal. It has improved my assessment of the paper, and I am increasing my score by one point.
> >
> > The additional evidence on constraint violation is helpful and makes it more convincing that the returned solution closely satisfies the fairness constraints in practice. I also appreciate the clarification that the optimization target and the reported balance metric are not identical, but that the latter is used for consistency with prior work. The discussion of convergence is still not a full formal guarantee for the proposed alternating scheme, but it gives a clearer justification for the algorithmic design and its relationship to existing nonconvex ADMM arguments.
> >
> > I still think some limitations remain, especially regarding the mismatch between optimized fairness and evaluated fairness, and the relatively easy synthetic quality benchmark. However, the rebuttal addresses a substantial part of my original concerns and makes the contribution more convincing overall.

---

### Official Review · Reviewer_t8pc · 2026-03-13

**Soundness:** 3
**Presentation:** 3
**Significance:** 3
**Originality:** 3
**Overall Recommendation:** 4
**Confidence:** 4

**Summary:**

This paper proposes a R-FairSC framework, which defines the fair spectral clustering problem as a constrained optimization problem on Riemannian manifold. This method decouples the manifold constraints and linear fairness constraints by directly encoding orthogonality constraints into the manifold geometry and introducing a variable splitting strategy into the ADMM framework, which enables scalable updates without expensive eigendecompositions. Experimental results on both synthetic and real-world datasets show that R-FairSC achieves a significant improvement in computational efficiency while maintaining clustering quality and fairness comparable to existing methods.

**Compliance With Llm Reviewing Policy:**

Affirmed.

**Ethical Review Flag:**

Flag this paper for an ethics review.

**Final Justification:**

Overall, this is a good paper. The rebuttal increased my confidence in the paper’s soundness and practical relevance without changing my original evaluation.

**Key Questions For Authors:**

1. This paper is currently entirely theoretical, lacking experimental results. Are there any experiments on synthetic or real-world datasets to demonstrate that the proposed algorithm outperforms previous research?

2. The algorithm relies on the "bounded displacement assumption." If some points violate this assumption, how will the approximation ratio change, will it completely fail?

3. (k, s)-spread is the core theoretical innovation of this paper. In practical applications, the global distribution of points is often unknown in advance. How to determine s beforehand?

**Limitations:**

yes

**Strengths And Weaknesses:**

The Strength can be summarized as follows.

Soundness: This paper is theoretically rigorous, with solid mathematical derivations and complexity analysis. It provides extensive experimental results, comparing against strong baselines such as FairSC, s-FairSC, and the latest A-FairSC, and evaluating performance on multiple metrics including error rate, balance, NCut, and runtime.

Presentation: This paper is clearly structured, starting with the limitations of existing methods, then introducing Riemannian manifolds, and then detailing the variable splitting strategy, making it easy for readers to follow.

Significance: The proposed method overcomes the traditional O(n3) time complexity limitation, improving computational efficiency. It also bridges the gap between Riemannian optimization and the fair spectral clustering problem.

Originality: This is the first work that applies Riemannian manifold optimization to the fair spectral clustering problem. It develops an efficient Riemannian ADMM algorithm that utilizes variable splitting to decouple constraints, enabling scalable updates without expensive eigensolvers.

The weakness can be summarized as follows.

Soundness: Section 3.3 mentions that the algorithm stops when the norm of the Riemannian gradient is less than ϵ or when the maximum number of iterations is reached. However, this paper lacks discussion on how to choose ϵ.

Presentation: Section 4.2 mentions that "R-FairSC remains highly competitive, yielding up to 70% lower NCut values than A-FairSC on the Bank and Credit datasets", but lacks further discussion on the reasons behind this phenomenon.

Significance: Aside from the improved running speed, in some cases (e.g., Figure 5), clustering quality such as NCut appears to be comparable to or slightly worse than the baseline.

Originality: The Riemann optimization framework can be directly applied to the k-way NCut problem. The originality of this paper lies mainly in its specific application to fair spectral clustering.

---

> ### Author Rebuttal · Authors · 2026-03-31
>
> We thank the reviewer for for the constructive feedback. Below we respond to the concerns raised.
>
> > #### W1. Regarding the stopping tolerance $\epsilon$ in Section 3.3.
>
> Thank you for pointing this out. In our experiments, we set $\epsilon = 10^{-4}$. We found that this value strikes an effective balance by ensuring stable convergence of the subproblem in practice while avoiding unnecessary computational overhead. Furthermore, $10^{-4}$ is a standard tolerance widely adopted in first-order Riemannian optimization methods to indicate practical stationarity. We agree that this detail is important for reproducibility, and we will explicitly state this parameter choice and our rationale in Section 3.3 of the revised manuscript.
>
> ---
>
> > #### W2. Section 4.2 mentions that "R-FairSC remains highly competitive, yielding up to 70\% lower NCut values than A-FairSC on the Bank and Credit datasets", but lacks further discussion on the reasons behind this phenomenon.
>
> Thank you for the comment. We believe that the observed performance gap is due to the distinct optimization formulations from which R-FairSC and A-FairSC are derived. Specifically, we reformulate fair spectral clustering as a constrained optimization problem over the Stiefel manifold. By introducing a variable-splitting strategy with the consensus constraint $H-Y=0$, we ensure that the fairness constraints remain strictly aligned with the original formulation $F^\top H=0$.
>
> In contrast, A-FairSC casts fair spectral clustering within a difference-of-convex (DC) framework, formulates the problem in terms of $M^2$ instead of $M$, and introduces a variable augmentation where the ADMM variable $Y$ is coupled with $MH$ rather than directly with $H$. As noted in their work, this design choice is motivated by efficient dualization and is intended to "effectively promote the same group balance." We believe these formulation differences inherently affect the optimization path and the resulting clustering quality. The difference in optimization behavior helps explain why R-FairSC attains lower NCut values than A-FairSC on the Bank and Credit datasets while maintaining competitive fairness. We will add a detailed discussion of these factors to Section 4.2 in the revised manuscript.
>
> ---
>
> > #### W3. Aside from the improved running speed, in some cases (e.g., Figure 5), clustering quality such as NCut appears to be comparable to or slightly worse than the baseline.
>
> Thank you for the observation. We would like to clarify that R-FairSC achieves average NCut values lower than or comparable to A-FairSC, the state-of-the-art baseline, in most cases (with the exception of the Diabetes dataset at $k=8$), while simultaneously achieving higher or comparable average fairness scores. Furthermore, these results are achieved at a fraction of the computational cost. Achieving comparable performance to state-of-the-art baselines while fundamentally resolving their computational bottlenecks is exactly the primary contribution and practical advantage of our approach.
>
> ---
>
> > #### W4. Regarding the originality of the paper.
>
> Thank you for the comment. We would like to emphasize that our contribution is not merely the direct application of Riemannian optimization to the standard k-way NCut problem. The main novelty of our work lies in how we formulate the fair spectral clustering problem to make it tractable. In the standard problem formulation, the variable $H$ must simultaneously satisfy two fundamentally different types of constraints: the orthogonality constraint and the linear fairness constraints. This coupled structure makes the problem difficult to solve directly. Our key idea is a variable-splitting approach that introduces an auxiliary variable $Y$ to handle the fairness constraints separately, while the main variable $H$ is subject only to the orthogonality constraint. This decoupling makes the resulting subproblems much easier to handle. The $H$-subproblem can be optimized efficiently on the Stiefel manifold using Riemannian optimization, while the $Y$-subproblem admits a closed-form solution. As a result, our method avoids the computationally expensive eigendecomposition required by prior fair spectral clustering methods, achieving superior computational efficiency while maintaining high clustering quality.
>
> ---
>
> > #### Regarding questions 1&ndash;3.
>
> Thank you for your questions. We believe there may be some confusion with another submission. The present paper already includes extensive experiments in Section 4 on both synthetic and real-world datasets, and evaluates clustering quality, fairness, and runtime against prior baselines. Moreover, our method is developed for fair spectral clustering through Riemannian optimization and ADMM frameworks. The notions of a bounded displacement assumption and $(k,s)$-spread do not appear in our formulation. We would therefore appreciate it if the reviewer could verify that these comments were intended for this submission.

---

> > ### Author Rebuttal · Reviewer_t8pc · 2026-04-03
> >
> > Thank you for the detailed response. I prefer to maintain my initial score at this stage.

---

### Official Review · Reviewer_CAMb · 2026-03-13

**Soundness:** 3
**Presentation:** 3
**Significance:** 2
**Originality:** 2
**Overall Recommendation:** 4
**Confidence:** 3

**Summary:**

This paper studies the Fair Spectral Clustering problem on graphs, which aims to minimize the normalized cut value subject to an additional linear constraint that enforces group fairness. The authors reformulate the problem as a Riemannian optimization problem and introduce an additional variable so that the final formulation is amenable to ADMM. The authors show that the ADMM subproblems can be solved efficiently either in closed-form or using a Riemannian Conjugate Gradient method. Empirically over both synthetic and real-world graphs, the authors show that the proposed method gains significant speed-up over existing baselines, while maintaining the same level of clustering quality.

**Compliance With Llm Reviewing Policy:**

Affirmed.

**Final Justification:**

Overall a good paper, would like to see it accepted. Not a higher score due to potentially limited impact.

**Key Questions For Authors:**

- I'm quite confused by the definition of balance in Equation (25). It is stated right after that "...1 denotes a perfectly balanced distribution and values near 0 indicate significant imbalance". The statement sounded like a score of 1 is the best whereas a score of 0 is the worst. However, since the global proportions may not be perfected balanced in the first place, e.g., when a sensitivity group has global proportion 0.01, a balance score of 1 for a particular cluster would mean a totally unfair clustering. You are reporting the average balance across all clusters, which should be fine. But such distinction should be more clearly explained.
- The Y-subproblem is modified so that a closed-form solution exists. Is this the reason to modify the ADMM subproblem? Does this modification affect the convergence of ADMM?

**Limitations:**

The authors did not discuss limitations, but I think that is fine for this work. The authors did discuss potential societal impact.

**Strengths And Weaknesses:**

Strengths:
- The problem is well-motivated and a technically sound solution is well presented.
- The numerical experiments are well executed. The plots and tables are informative and clearly illustrate the improvement over existing baselines.

Weaknesses:
- (*I'm not sure if this should be a weakness or a strength*) The technical component of this work is pretty simple and straightforward.
- (*Minor*) When citing references, please use \citet and \citep appropriately, e.g. in Line 87 (col 2), Line 119 (col 1), Line 151 (col 1), Line 324 (col 2).
- (*Typo*) I think the rhs of Equation (19) is missing H_l in the parenthesis.

---

> ### Author Rebuttal · Authors · 2026-03-31
>
> We thank the reviewer for the constructive feedback. Below we respond to the concerns raised.
>
> > #### W1. The technical component of this work is pretty simple and straightforward.
>
> Thank you for the comment. While the final algorithm is straightforward to implement, the formulation required to achieve this simplicity is highly non-trivial. The main novelty of our work lies in how we formulate the fair spectral clustering problem to make it tractable. In the original problem, the variable $H$ must simultaneously satisfy two different types of constraints: the orthogonality constraint and the linear fairness constraints. This coupled structure makes the problem difficult to solve directly. Our key idea is a variable-splitting approach that introduces an auxiliary variable $Y$ to handle the fairness constraints separately, while the main variable $H$ is subject only to the orthogonality constraint. This decoupling makes the resulting subproblems much easier to handle. The $H$-subproblem can be optimized efficiently on the Stiefel manifold using Riemannian optimization, while the $Y$-subproblem admits a closed-form solution. As a result, our method avoids the computationally expensive eigendecomposition required by prior fair spectral clustering methods, achieving superior computational efficiency while maintaining high clustering quality.
>
> ---
>
> > #### W2. When citing references, please use \citet{} and \citep{}, e.g. in Line 87 (col 2), Line 119 (col 1), Line 151 (col 1), Line 324 (col 2).
>
> Thank you for pointing this out. We will ensure all citations appear correctly by using \citet{} and \citep{} instead of \cite{} throughout the revised manuscript.
>
> ---
>
> > #### W3. I think the rhs of Equation (19) is missing $H_l$ in the parenthesis.
>
> Thank you for your careful reading. We would like to clarify that the right-hand side of Eq. (19) is indeed a function of $H_\ell$. Specifically, $d_\ell$ denotes the Riemannian conjugate gradient search direction at the current inner iteration $\ell$. As indicated at Line 272, this is a tangent vector associated with the current iterate $H_\ell$. Furthermore, the retraction $R_{H_\ell}(\alpha_\ell d_\ell)$ inherently operates on the current iterate $H_\ell$, utilizing the QR-based retraction defined as $R_{H_\ell}(\alpha_\ell d_\ell)=qf(H_\ell+\alpha_\ell d_\ell)$.
>
> ---
>
> > #### Q1. Regarding the definition and interpretation of the balance metric.
>
> Thank you for pointing this out. It is correct that a balance score of 1 implies equal representation within a cluster, which may differ significantly from the global proportions if the dataset is inherently skewed. To clarify, the metric in Eq. (25) evaluates absolute parity among demographic groups within the same cluster (i.e., the minimum pairwise ratio of their representations), rather than proportional representation relative to the global dataset. Thus, "perfectly balanced" here refers strictly to equal representation within a specific cluster. As you correctly noted, taking the average across all clusters ensures we are evaluating fairness globally across the entire partition. We also note that this metric has been widely used in prior studies on fair spectral clustering. In the revised manuscript, we will explicitly clarify this distinction between absolute within-cluster parity and global proportional representation to avoid any confusion.
>
> ---
>
> > #### Q2. The Y-subproblem is modified so that a closed-form solution exists. Is this the reason to modify the ADMM subproblem? Does this modification affect the convergence of ADMM?
>
> Thank you for the question. We would like to clarify that the $Y$-subproblem is not modified simply to obtain a closed-form solution. Rather, it naturally arises from our variable-splitting reformulation, which is designed to separate the fairness constraints from the orthogonality constraint so that each subproblem can be solved efficiently. Specifically, we show that the $Y$-subproblem reduces to the Euclidean projection onto the linear subspace $\mathrm{null}(F^\top)$, which admits a closed-form solution. Regarding the effect of this reformulation on convergence, we kindly refer the reviewer to our response to W3 for Reviewer rPpY, where we provide a detailed discussion of the convergence characteristics of R-FairSC.

---

> > ### Author Rebuttal · Reviewer_CAMb · 2026-04-02
> >
> > Thank you for the detailed response! Please include a discussion on the convergence of the algorithm in the revised version.

---

### Decision · Program_Chairs · 2026-04-30

**Decision:**

Accept (regular)

**Comment:**

The paper presents a scalable solution to fair clustering. Reviewers were overall moderately enthusiastic about the paper, highlighting its clear motivation, soundness of the formulation and algorithm, and clarity of the writing.
The evaluation is comprehensive, following proper work for synthetic evaluation and testing on large-scale real-world data (tens of thousands of nodes, hundreds of thousands to several million edges), comparing to appropriate baselines. The method has moderate (x5) to significantly improved runtime compared to competing methods with better or comparable performance.

The authors should take the reviewer feedback into account for the final version of the paper, specifically regarding convergence, experiments on the constraint violation constraint and adding error bars to all plots. As a minor comment $f(H)$ is defined rather late in the paper.